# Molecular Insights into Innate Immune Response in Captive Koala Peripheral Blood Mononuclear Cells Co-Infected with Multiple Koala Retrovirus Subtypes

**DOI:** 10.3390/pathogens11080911

**Published:** 2022-08-14

**Authors:** Mohammad Enamul Hoque Kayesh, Md Abul Hashem, Fumie Maetani, Atsushi Goto, Noriko Nagata, Aki Kasori, Tetsuya Imanishi, Kyoko Tsukiyama-Kohara

**Affiliations:** 1Transboundary Animal Diseases Centre, Joint Faculty of Veterinary Medicine, Kagoshima University, Kagoshima 890-0065, Japan; 2Department of Microbiology and Public Health, Faculty of Animal Science and Veterinary Medicine, Patuakhali Science and Technology University, Barishal 8210, Bangladesh; 3Hirakawa Zoological Park, Kagoshima 891-0133, Japan; 4Awaji Farm Park England Hill Zoo, Minamiawaji 665-0443, Japan; 5Tama Zoological Park, Tokyo 191-0041, Japan; 6Kanazawa Zoo, Yokohama 236-0042, Japan; 7Nagoya Higashiyama Zoo, Nagoya 464-0804, Japan

**Keywords:** koala, peripheral blood mononuclear cells, innate immune response, TLRs, cytokines, koala retrovirus

## Abstract

Koala retrovirus (KoRV) exists in both endogenous and exogenous forms and has appeared as a major threat to koala health and conservation. Currently, there are twelve identified KoRV subtypes: an endogenous subtype (KoRV-A) and eleven exogenous subtypes (KoRV-B to -I, KoRV-K, -L, and -M). However, information about subtype-related immune responses in koalas against multiple KoRV infections is limited. In this study, we investigated KoRV-subtype (A, B, C, D, and F)-related immunophenotypic changes, including CD4, CD8b, IFN-γ, IL-6, and IL-10 mRNA expression, in peripheral blood mononuclear cells (PBMCs) obtained from captive koalas (n = 37) infected with multiple KoRV subtypes (KoRV-A to F) reared in seven Japanese zoos. Based on KoRV subtype infection profiles, no significant difference in CD4 and CD8b mRNA expression was observed in the study populations. Based on the different KoRV subtype infections, we found that the IFN-γ mRNA expression in koala PMBCs differs insignificantly (*p* = 0.0534). In addition, IL-6 and IL-10 mRNA expression also did not vary significantly in koala PBMCs based on KoRV subtype differences. We also investigated the Toll-like receptors (TLRs) response, including TLR2–10, and TLR13 mRNA in koala PBMCs infected with multiple KoRV subtypes. Significant differential expression of TLR5, 7, 9, 10, and 13 mRNA was observed in the PBMCs from koalas infected with different KoRV subtypes. Therefore, based on the findings of this study, it is assumed that co-infection of multiple KoRV subtypes might modify the host innate immune response, including IFN-γ and TLRs responses. However, to have a more clear understanding regarding the effect of multiple KoRV subtypes on host cytokines and TLR response and pathogenesis, further large-scale studies including the koalas negative for KoRV and koalas infected with other KoRV subtypes (KoRV-A to -I, KoRV-K, -L and -M) are required.

## 1. Introduction

Koala retrovirus (KoRV) is a recently discovered gammaretrovirus found in koalas (*Phascolarctos cinereus*) with leukemia, lymphoma, and immunodeficiency-like diseases [1]. KoRV belongs to the family Retroviridae and has a positive-sense, single-stranded RNA genome of approximately 8.4 kb, containing the gag, pol, and env genes. An integrated KoRV provirus additionally contains long terminal repeats (LTRs) at the 5′ and 3′ ends [1,2,3]. KoRV is a source of great concern for conservationists and also a source of great interest for virologists, immunologists, and epidemiologists, as it is one of the only viruses existing in both exogenous and endogenous forms. Moreover, KoRV infections and manmade hazards have caused a severe decline in koala populations [4,5], resulting in the koala being listed as vulnerable in the ‘red list’ of threatened species of International Union for Conservation of Nature [6]. However, authorities in Australia are now reviewing their survival status with the aim to re-classify the species as endangered due to its rapid decline, where KoRV infection in koalas remains as one of the major health problems and poses an additional challenge for koala conservation [4,7]. KoRV has an interesting feature of currently existing in both endogenous and exogenous forms [1,8,9], which has drawn the attention of many virologists and immunologists across the world [10].

So far, twelve KoRV subtypes (KoRV-A to -I, KoRV-K, -L, and -M) have been reported [11,12]. KoRV-A is considered an endogenous subtype, found in all the tested northern koala populations with habitats in Queensland, and spreading to the southern koala populations with habitats in Victoria and South Australia [7]. Other subtypes (KoRV-B to -I and KoRV-K, -L, and -M) are considered to be exogenous [7,11,12]. Many recombinant KoRVs (recKoRV) have also recently been reported in koalas [13,14,15]. The recombination between endogenous retrovirus (ERV) and exogenous retrovirus (XRV) may generate ERV-XRV chimeras [16]. Although the direct causation of disease is currently unclear, KoRV infection has been suspected to cause immunosuppression in koalas, which may result in opportunistic infections, including chlamydiosis [17,18,19]. 

CD4 and CD8 are important coreceptor molecules expressed on T helper cells and cytotoxic T lymphocytes, respectively, and are used to enhance responses mediated via the TCR [20,21,22]. The innate immune response is essential for controlling viral infections and shaping adaptive immunity [23,24]. Toll-like receptors (TLRs) are key components of innate immunity and recognize conserved microbial structures and pathogen-associated molecular patterns. TLRs are type I transmembrane proteins with an N-terminal ectodomain consisting of leucine-rich repeats, a single transmembrane domain, and a cytosolic TIR domain [25]. TLR activation, in turn, induces interferons (IFNs), cytokines, and chemokines via several distinct signaling pathways, thus promoting adaptive immune responses and limiting infection [26,27]. However, there is currently a gap in our understanding of the host innate immune response to endogenous and exogenous KoRVs in koalas [10]. Previous studies have identified ten TLRs, including TLR2–10 and TLR13, in koalas [28]. However, the role of TLRs in host immune response to KoRV infection with multiple subtypes remains unclear [10]. In our previous work, we characterized the host innate immune response in a small number of koalas (n = 10 or 11) infected with a range of KoRV subtypes (KoRV-A only vs. KoRV-A with KoRV-B and/or -C) and determined the TLR and cytokine expression, including IL-6, IL-10, and IL-17A, in PBMCs [29,30]. Further elucidation of these response pathways is critical for successful preventive and therapeutic interventions. Therefore, in this study, we aimed to characterize the zoo-dwelling koala population in Japan to extend our insights into host innate immune response in koala PBMCs against multiple KoRV subtypes infection. To this end, we investigated the expression patterns of immune molecules CD4 and CD8b, cytokines (IFN-γ, IL-6, and IL-10), and TLRs (TLR2–10, and TLR13) mRNAs in a larger number of captive koala populations (n = 37), based on KoRV subtype profiles, including KoRV-A, KoRV-A+B, KoRV-A+B+D, KoRV-A+B+F, KoRV-A+D+F, KoRV-A+B+C+D, KoRV-A+B+D+F, and KoRV-A+B+C+D+F infections [31].

## 2. Materials and Methods

### 2.1. Animals

In this study, we targeted captive koalas (n = 37) dwelling in seven different Japanese zoos, including Kanazawa Zoo, Tama Zoological Park, Kobe Oji Zoo, Awaji Farm England Hill Zoo, Hirakawa Zoological Park, Saitama Children’s Zoo, and Nagoya Higashiyama Zoo, and these represent all seven zoos in Japan which rear koalas. In all seven zoos, the koalas were maintained in an air conditioning system (23~25 °C). In addition, as much as possible, each habitat is designed to simulate the koalas’ natural habitat, and koalas were provided with an ad libitum supply of eucalyptus leaves. These koalas were available for blood sampling one time during the study period (June to October 2021) for the characterization of the innate immune response against multiple KoRV subtypes infections. From the captive koalas, whole blood samples were obtained by venipuncture using EDTA/heparin between June and October 2021. The koalas used in this study have previously been investigated for multiple KoRV subtype infections, including KoRV-A to -F [31]. Figure 1 illustrates the workflow followed in this study. This study was performed in accordance with the protocols of the Institutional Animal Care and Use Committee of the Joint Faculty of Veterinary Medicine, Kagoshima University, Japan.

### 2.2. Hematological Examination and Clinical Sign Observation

Hematological examination was performed following standard protocols to determine white blood cell (WBC) count. Clinical signs of each koala used in this study were observed by trained zoo veterinarians to understand the health status of koalas. 

### 2.3. Extraction of Genomic DNA

Genomic DNA (gDNA) was extracted from whole blood (300 µL) using a Wizard Genomic DNA Purification Kit (Promega), according to the manufacturer’s instructions. The purity and concentration of the extracted DNA were measured using a NanoDrop ND-1000 spectrophotometer (NanoDrop Technologies, Inc., Waltham, MA USA). Extracted gDNA was used as a template for KoRV provirus determination by PCR analysis, and also for KoRV subtyping. KoRV provirus was confirmed by PCR analysis, using primers targeting KoRV pol gene, pol F (5′-CCTTGGACCACCAAGAGACTTTTGA-3′) and pol R (5′-TCAAATCTTGGACTGGCCGA-3′), as described previously [3]. For KoRV subtyping, a PCR was performed to detect KoRV subtypes, using subtype-specific primers targeting the env gene in gDNA. The primers used for detecting KoRV-A to -F subtypes are described in our previous study [31]. The infection status of KoRV-A, KoRV-B, KoRV-C, KoRV-D, KoRV-E, and KoRV-F subtypes in the study populations was confirmed in our previous study [31]. The infection status of each animal for the KoRV subtype was confirmed by genotyping PCR and sequencing, and the extracted gDNA was used as template for PCR. 

### 2.4. Extraction of RNA from Koala Peripheral Blood Mononuclear Cells (PBMCs)

PBMCs were isolated from whole blood samples obtained from the koalas, as reported previously [32] but with a few modifications [33]. Total RNA was extracted from the isolated PBMCs using the RNeasy Plus Mini Kit (QIAGEN, Hilden, Germany) according to the manufacturer’s instructions. The purity and concentration of the extracted RNA were determined using a NanoDrop ND-1000 spectrophotometer (NanoDrop Technologies Inc., Waltham, MA, USA). The extracted RNA samples were stored at −80 °C until further use for gene expression analysis.

### 2.5. Gene Expression Analysis by Quantitative Reverse Transcription-PCR (qRT-PCR)

To investigate the mRNA expression patterns of CD4, CD8b, cytokines (IFN-γ, IL-6, and IL-10), and TLRs (TLR2–10 and TLR13) in koala PBMCs, a one-step qRT-PCR was performed, as described previously [29,30]. The reaction condition for measuring the IFN-γ mRNA expression was similar to other cytokines, as described previously [29]. Primers used in this study are shown in Table 1.

### 2.6. Statistical Analysis

A one-way ANOVA was performed using GraphPad Prism (Prism 9 for macOS, by Dennis Radushev, San Diego, CA, USA) to compare gene expression among different groups based on the KoRV subtype infection profiles, including KoRV-A, KoRV-A+B, KoRV-A+B+D, KoRV-A+B+F, KoRV-A+D+F, KoRV-A+B+C+D, KoRV-A+B+D+F, and KoRV-A+B+C+D+F. A Mann–Whitney U test was performed using GraphPad Prism (version 9) software to compare gene expression difference between the groups. Statistical significance was set at *p* < 0.05.

## 3. Results

### 3.1. Infection Status of Koalas for KoRV Subtypes

In this study, 37 captive koalas, including 13 male koalas between the ages of 11 months and 12 years (median age: 4 years 6 months) and 24 female koalas between 10 months and 24 years (median age: 5 years 7 months) across seven zoos in Japan, were used. Using KoRV pol gene-specific primers, PCR analysis of the gDNA extracted from whole blood showed that all the koalas were positive for the KoRV provirus. The infection status of the individual koalas for the KoRV subtypes, including KoRV-A, KoRV-B, KoRV-C, KoRV-D, KoRV-E, and KoRV-F, was determined in a previous study and confirmed by genotyping PCR and sequencing [31]. Data on the age, sex, and infection status of the individual koalas with KoRV subtypes are presented in Table 2. The previous study demonstrated that there were no significant differences in the KoRV proviral and KoRV RNA loads based on the infection subtype profiles [31]. Of the 37 koalas, 34 showed no abnormal clinical signs; however, two koalas presented with loose stools and one with abdominal bloating [31]. The WBC counts of all the koalas were within the normal range, and the koalas were defined as healthy (Table 2).

### 3.2. Expression Patterns of CD4 and CD8b mRNA in Koala PBMCs

To investigate the CD4 and CD8b mRNA expressions in koala PBMCs infected with multiple subtypes, we measured the expression levels of these molecules by qRT-PCR. The expression levels of CD4 and CD8b mRNA were normalized against koala beta actin mRNA levels. In one-way ANOVA analysis among the groups, CD4 and CD8b showed no significant difference based on infection subtype profiles (Figure 2A,B).

### 3.3. Expression Patterns of Cytokine mRNAs in Koala PBMCs

To investigate the cytokine mRNAs, including IFN-γ, IL-6, and IL-10 mRNA expression in koala PBMCs infected with multiple subtypes, we measured the expression levels of these molecules by qRT-PCR. The expression levels of IFN-γ, IL-6, and IL-10 mRNA were normalized against koala beta actin mRNA levels. For cytokine expression analysis among the groups, a one-way ANOVA was performed. Based on the different KoRV subtype infections, IFN-γ mRNA expression was found to be varied; however, it was not significant (*p* = 0.0534) (Figure 3A). IL-6 and IL-10 mRNA expression also did not significantly differ (Figure 3B,C).

### 3.4. Expression Patterns of TLR mRNAs in Koala PBMCs

To investigate immune response based on KoRV subtype differences, we measured the expression patterns of TLR (TLR2–10 and TLR13) mRNAs in koala PBMCs by qRT-PCR. It was found that TLR2–10 and TLR13 were expressed at the mRNA level in koala PBMCs (Figure 4A–J). One-way ANOVA revealed a significantly different expression of TLR5, 7, 9, 10, and 13 mRNA across koala PBMCs infected with multiple KoRV subtypes, including KoRV-A, KoRV-A+B, KoRV-A+B+D, KoRV-A+B+F, KoRV-A+D+F, KoRV-A+B+C+D, KoRV-A+B+D+F, and KoRV-A+B+C+D+F (Figure 4D,F,H–J). However, no significant differences were found in the TLR2, TLR3, TLR4, TLR6, and TLR8 mRNA expressions (Figure 4A–C,E,G).

### 3.5. Difference in the Expression Patterns of CD4, CD8b, Cytokines, and TLR mRNAs between the Groups in Koala PBMCs

We also performed analysis between the groups such as KoRV-A+B vs. KoRV-A+B+D, KoRV-A+B vs. KoRV-A+B+F, KoRV-A+B+D vs. KoRV-A+B+C+D, KoRV-A+B+D vs. KoRV-A+B+D+F, and KoRV-A+B+D+F vs. KoRV-A+B+C+D+F using the Mann–Whitney U test to determine the effect of individual serotypes on CD4, CD8b, IFN-γ, IL-6, IL-10, and TLRs (TLR2–10 and TLR13) mRNA expressions in koala PBMCs. When comparing the KoRV-A+B and KoRV-A+B+D-infected groups, significant difference in the mRNA expression of IFN-γ, TLR3, TLR4, TLR7, and TLR10 was observed (Appendix A). However, no significant difference was observed in CD4, CD8b, IL-6, IL-10, TLR2, TLR6, TLR8, TLR9, and TLR13 mRNA expressions (Appendix A).

When comparing the KoRV-A+B and KoRV-A+B+F-infected groups, only TLR6 mRNA expression showed a significant difference, and no other molecules showed any significant difference (Appendix A).

When comparing the KoRV-A+B+D and KoRV-A+B+C+D-infected groups, IFN-γ, IL-6, TLR5, TLR7, and TLR10 mRNA expressions showed significant differences, and no other molecules showed any significant difference (Appendix A).

When comparing the KoRV-A+B+D and KoRV-A+B+D+F-infected groups, TLR3, TLR5, and TLR10 mRNA expressions showed significant differences, and no other molecules showed any significant difference (Appendix A).

When comparing the KoRV-A+B+D+F and KoRV-A+B+C+D+F-infected groups, no molecules showed any significant difference in gene expression patterns (Appendix A).

## 4. Discussion

Until recently, 12 KoRV subtypes are identified [11,12], which might impact koala health and its conservation by exerting effects on host innate immune response. Therefore, in this study, we profiled the expression patterns of important coreceptor molecules such as CD4 and CD8b, cytokines, and TLR molecules. To the best of the authors’ knowledge, this is the first study to profile the innate immune response of koala PBMCs infected with KoRV-A+B+D, KoRV-A+B+F, KoRV-A+B+C+D, KoRV-A+B+D+F, and KoRV-A+B+C+D+F subtypes.

This study will also extend the understanding of our recently published data [29], where we characterized the expression patterns of CD4, CD8b, and cytokines in koala PBMCs infected with either KoRV-A or KoRV-A+B or KoRV-A+B+C, obtained from 10 captive koalas reared in Hirakawa Zoological Park, Kagoshima, Japan. Additionally, the findings of this study will also extend the understanding of our recently published study of TLR response in koala PBMCs infected with either KoRV-A or KoRV-A+B or KoRV-A+B+C in 11 captive koala populations obtained from two Japanese zoos (Hirakawa Zoological Park, Kagoshima, Japan, and Kobe Oji Zoo, Kobe, Japan) [30]. Notably, in this study, we have covered all seven zoos exhibiting koalas in Japan. However, we had some limitations, including the absence of KoRV-negative koalas in this study, of which inclusion might provide more a clear understanding regarding the influence of KoRV infection on koalas’ immune responses. The prevalence of each serotype (KoRV-A to -F) of these samples has been indicated, where KoRV-B was the most prevalent non-KoRV-A subtype in captive populations [31], which differs from the recently reported prevalence of these subtypes in wild northern koalas, where KoRV-D has been reported as the most prevalent non-KoRV-A subtype [11].

In this study, the difference in CD4 and CD8b expression among different groups of multiple KoRV subtypes was not found to be significant in these koala populations. However, it was found that the presence of KoRV-C might influence a decrease in CD4 mRNA expression (Figure 2A). The same pattern was also observed in IFN-γ mRNA expression (Figure 3A). A large-scale investigation is required for the further confirmation of the potential for KoRV-C to cause a decrease in CD4 and IFN-γ mRNA expression.

Significant differences in the expression of TLR5, TLR7, TLR9, TLR10, and TLR13 mRNAs were observed, suggesting the effect of different KoRV subtypes on TLR expression. We also cannot rule out the possibility of the presence of genetically susceptible koalas, which would influence the host–virus interaction modulating immune response. However, the findings of the present study indicate that there are different expressions of TLRs, CD4, CD8b, and cytokines in these captive koala populations, which might be associated with different KoRV subtype infections. However, further large-scale studies including the koalas negative for KoRV and koalas infected with other KoRV subtypes (KoRV-A to -I, KoRV-K, -L and -M) are required to gain a deeper understanding regarding the effect of KoRV subtypes on host immune response as well as koala health and conservation.

## Figures and Tables

**Figure 1 pathogens-11-00911-f001:**
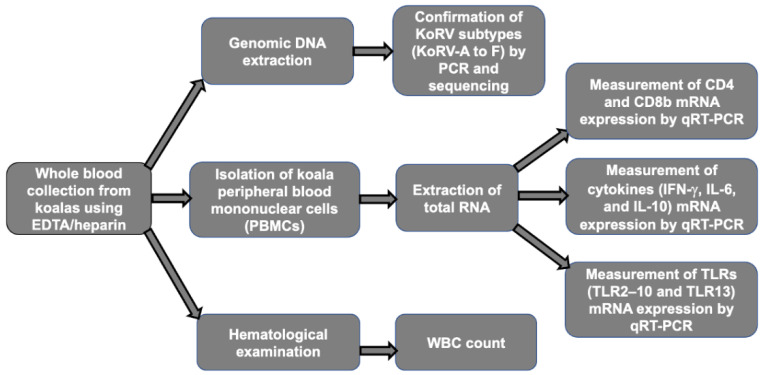
A schematic representation of the workflow of the study. Whole blood sample was collected from koalas using EDTA/heparin, which was subjected to gDNA extraction, PBMCs isolation, and for WBC count. The extracted gDNA was used for KoRV provirus detection and KoRV subtyping, and the extracted RNA from PBMCs was used for gene expression analysis.

**Figure 2 pathogens-11-00911-f002:**
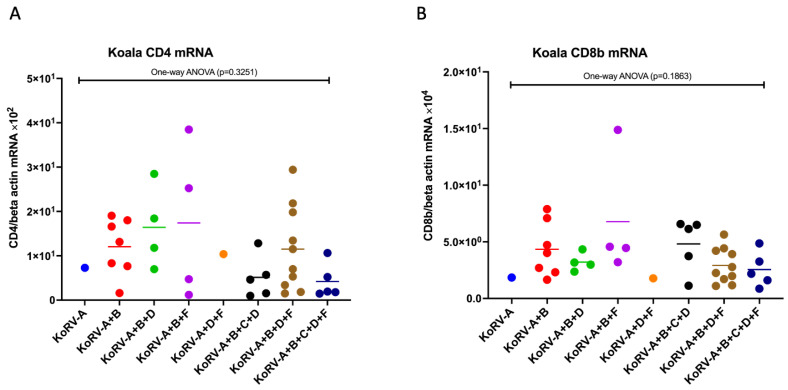
CD4 and CD8b mRNA expression levels in koala peripheral blood mononuclear cells (PBMCs) infected with different KoRV subtypes. mRNA expressions of CD4 (**A**) and CD8b (**B**) are indicated in koala PBMCs based on the infection profiles of KoRV subtypes, including KoRV-A, KoRV-A+B, KoRV-A+B+D, KoRV-A+B+F, KoRV-A+D+F, KoRV-A+B+C+D, KoRV-A+B+D+F, and KoRV-A+B+C+D+F positivity. The transcript levels were normalized against koala beta actin. Statistical significance of CD4 and CD8b was evaluated using one-way ANOVA test among the groups and was found to be insignificant. Thick horizontal lines indicate arithmetic mean.

**Figure 3 pathogens-11-00911-f003:**
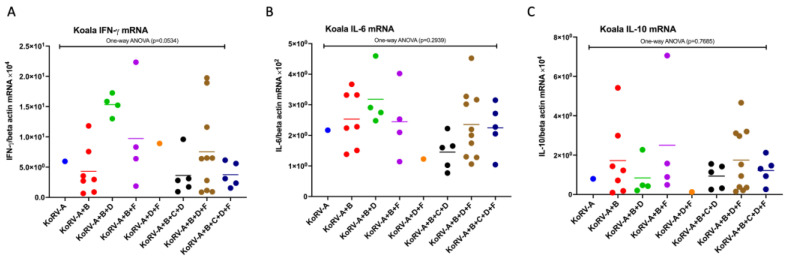
Cytokine mRNA expression level in koalas (PBMCs) infected with different KoRV subtypes. mRNA expressions of IFN-γ (**A**), IL-6 (**B**), and IL-10 (**C**) are indicated in koala PBMCs based on the infection profiles of KoRV subtypes, including KoRV-A, KoRV-A+B, KoRV-A+B+D, KoRV-A+B+F, KoRV-A+D+F, KoRV-A+B+C+D, KoRV-A+B+D+F, and KoRV-A+B+C+D+F positivity. The cytokine transcript levels were normalized against that of koala beta-actin. Statistical significance of IFN-γ, IL-6, and IL-10 mRNA expression was evaluated using a one-way ANOVA test among the groups and was found to be insignificant. Thick horizontal lines indicate arithmetic mean.

**Figure 4 pathogens-11-00911-f004:**
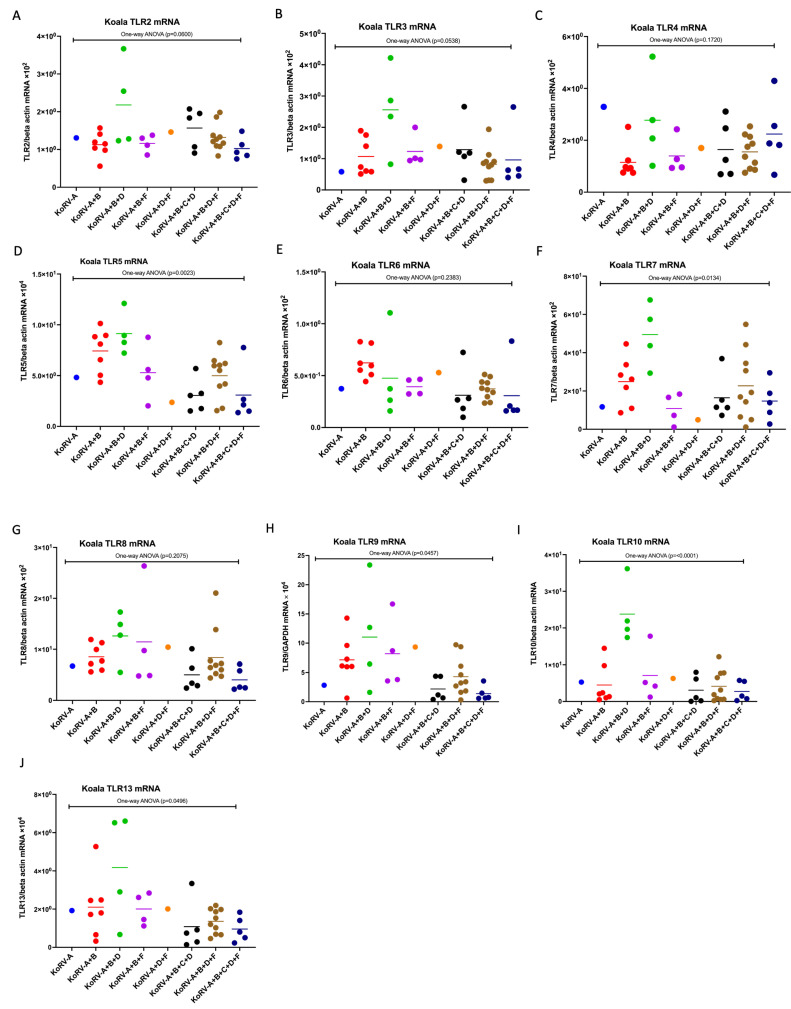
Expression profile of TLR mRNAs in koala PBMCs infected with different KoRV subtypes. Expression patterns of TLR2 (**A**), TLR3 (**B**), TLR4 (**C**), TLR5 (**D**), TLR6 (**E**), TLR7 (**F**), TLR8 (**G**), TLR9 (**H**), TLR10 (**I**), and TLR13 (**J**) mRNAs are indicated in koala PBMCs based on the infection profiles of KoRV subtypes, including KoRV-A, KoRV-A+B, KoRV-A+B+D, KoRV-A+B+F, KoRV-A+D+F, KoRV-A+B+C+D, KoRV-A+B+D+F, and KoRV-A+B+C+D+F positivity. The transcript levels were normalized against koala beta actin mRNA levels. A one-way ANOVA test was performed to analyze the significant difference in TLR response among the groups and has been indicated. Thick horizontal lines indicate arithmetic mean.

**Table 1 pathogens-11-00911-t001:** Primers used in this study.

Koala Gene	Forward (5′ to 3′)	Reverse (5′ to 3′)	Product Length (bp)	Reference
*TLR2*	CCATTCCAAGTGAGGGGCAA	ACTCCAGTCAGCAAGGCAAG	122	[30]
*TLR3*	GGAATGGCTTGGGTTGGAGT	AGCCACTGGAAAGAAAAATCATCT	162	[30]
*TLR4*	TCCACAAGAGCCGGAAAGTC	GAGTTCCACCTGTTGCCGTA	176	[30]
*TLR5*	CCTTAGCCTGGATGGCAACA	GGTAGGGTCAGGGGATAGCA	109	[30]
*TLR6*	TTCAGTTTCCCGCCCAACTA	ATGTGGCCATCCACTTACCA	157	[30]
*TLR7*	TTGCCTTGTAACGTCACCCA	GTGAGGGTCAGGTTGGTTGT	119	[30]
*TLR8*	CCTCTTCGTTTACCACCCTCC	CTTCAAAGGCCCCGTCATCT	178	[30]
*TLR9*	ATCTTCAGCCACTTCCGCTC	AGGCTCTCTCCAGCCCTAAA	133	[30]
*TLR10*	GCCCTAAAGGTGGAGCATGT	TATATGTGGCATCCCCGCAC	123	[30]
*TLR13*	AGCCTACTGGTGGCTATGGA	TGGCCAGGTACAGGGACTTA	172	[30]
*CD4*	GCCAACCCAAGTGACTCTGT	TCTCCTGGACCACTCCATTC	105	[34]
*CD8b*	GCATTGGCTTCTAATTGCTAGTATC	CACTTTCTATCATGCAAAGTAACCC	88	[29]
*IFN-γ*	CTGCCTGGTTACCTTCTTGCT	AACCCAACATAACACAAAGCCA	79	This study
*IL-6*	TGGATGAGCTGAACTGTACCC	GCTTGCCAAGGATTGTGAGT	119	[34]
*IL-10*	ACCAGAGACAAGCTCGAAAC	TCTTCCAGCAAAGATTTGTCTATC	50	[29]
*Beta actin*	AGATCATTGCCCCACCT	TGGAAGGCCCAGATTC	123	[3]

**Table 2 pathogens-11-00911-t002:** Health information details of the koalas used in this study.

Zoo	Koala	Age (During Sampling)	Sex	KoRV Subtypes						WBC (×10^2^/μL Blood)	Health Status
				KoRV-A	KoRV-B	KoRV-C	KoRV-D	KoRV-E	KoRV-F		
Kanazawa Zoo (n = 4)	KU_KAZ_01	1 yr 3 mo	F	P	P	N	N	ND	P	115	Healthy
	KU_KAZ_02	6 yr 8 mo	M	P	P	N	N	ND	P	75.5	Healthy
	KU_KAZ_3	8 yr 1 mo	F	P	P	P	P	ND	N	66.5	Healthy
	KU_KAZ_4	4 yr 3 mo	F	P	P	N	P	ND	P	95.5	Healthy
Tama Zoological Park (n = 2)	KU_TZ_05	6 yr	M	P	P	P	P	ND	N	NA	Healthy
	KU_TZ_06	5 yr	M	P	P	N	P	ND	P	NA	Healthy
Kobe Oji Zoo (n = 8)	KU_KZ_07	11 yr	F	P	P	N	N	ND	P	45	Healthy
	KU_KZ_08	7 yr 2 mo	F	P	P	N	P	ND	P	54	Healthy
	KU_KZ_09	5 yr 5 mo	M	P	P	N	P	ND	N	68	Healthy
	KU_KZ_10	4 yr 11 mo	F	P	N	N	P	ND	P	68.5	Healthy
	KU_KZ_11	4 yr	M	P	P	P	P	ND	P	30	Healthy
	KU_KZ_12	2 yr 8 mo	F	P	P	N	P	ND	P	147	Healthy
	KU_KZ_13	2 yr 3 mo	F	P	P	N	N	ND	N	NA	Healthy
	KU_KZ_14	2 yr 3 mo	F	P	P	N	N	ND	N	95.5	Healthy
Awaji Farm England Hill Zoo (n = 4)	KU_AZ_15	24 yr	F	P	N	N	N	ND	N	50	Healthy
	KU_AZ_16	13 yr	F	P	P	N	N	ND	N	58	Healthy
	KU_AZ_17	12 yr	M	P	P	N	P	ND	N	97	Healthy
	KU_AZ_18	7 yr	M	P	P		P	ND	P	42	Healthy
Hirakawa Zoological Park (n = 4)	KU_HZ_19	1 yr	M	P	P	P	P	ND	N	62	Healthy
	KU_HZ_20	1 yr	M	P	P	N	P	ND	N	50	Healthy
	KU_HZ_21	1 yr	F	P	P	N	P	ND	P	82	Healthy
	KU_HZ_22	1 yr	F	P	P	N	P	ND	N	88	Healthy
Saitama Children’s Zoo (n = 5)	KU_SZ_23	3 yr	M	P	P	N	P	ND	P	NA	Healthy
	KU_SZ_24	7 yr	F	P	P	P	P	ND	P	NA	Healthy
	KU_SZ_25	3 yr	F	P	P	P	P	ND	P	NA	Healthy
	KU_SZ_26	2 yr	F	P	P	P	P	ND	N	NA	Healthy
	KU_SZ_27	2 yr	F	P	P	P	P	ND	P	NA	Healthy
Nagoya Higashiyama Zoo (n = 10)	KU_NZ_28	9 yr	M	P	P	P	P	ND	N	76	Healthy
	KU_NZ_29	4 yr	M	P	P	N	P	ND	P	85	Healthy
	KU_NZ_30	11 yr	F	P	P	N	P	ND	P	23	Healthy
	KU_NZ_31	11 yr	F	P	P	P	P	ND	P	51	Healthy
	KU_NZ_32	7 yr	F	P	P	N	N	ND	N	47	Healthy
	KU_NZ_33	4 yr	F	P	P	N	P	ND	P	57	Healthy
	KU_NZ_34	10 mo	F	P	P	N	N	ND	N	43	Healthy
	KU_NZ_35	1 yr	F	P	P	N	N	ND	P	69	Healthy
	KU_NZ_36	11 mo	M	P	P	N	N	ND	N	69	Healthy
	KU_NZ_37	3 yr	F	P	P	N	N	ND	N	66	Healthy

P, positive; N, negative; M, male; F, female; yr, year; mo, month; ND, not detected; WBC, white blood cell; NA, data not available.

## Data Availability

The datasets generated during and/or analyzed during the current study can be find in the main text and the Appendix A.

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
