# Peer review of "Molecular Insights into Innate Immune Response in Captive Koala Peripheral Blood Mononuclear Cells Co-Infected with Multiple Koala Retrovirus Subtypes"

_pathogens, 2022, doi:10.3390/pathogens11080911_

Round 1
Reviewer 1 Report
This is an interesting study that I was pleased to review, as I believe the assessment of the immune response in koalas and wildlife species in general is still a challenge. Thus, contributions to this matter are extremely relevant for the better understanding of the response of these species to pathogens, especially considering their role in the one health context and their often endangered status.
Below are some comments and points I believe need further clarification.
· Is there any reason for not describing the recently identified KoRV subtypes L-M? I understand they wouldn’t be included in the study, but they are also not raised in the introduction.
· Line 23: IFN-γ
· Lines 48-50: I think this sentence is not really clear. I suggest rewriting it.
· Lines 58-59: What do Northern and Southern refer to? Australia?
· The inclusion in methods and results of lots of details that have been already published might not be needed or appropriate. I understand this information needs to be given, but the details could be more concise.
· In the methods, why did the authors decide to use PBMS instead of whole blood for the molecular study of immunological markers?
· Lines 161-162: Could you provide the median of the ages?
· Line 163: where it says PBMC, isn’t it whole blood?
· Line 192: KoRV-A is listed amongst the subtypes but haven’t been analysed.
· Line 197: Instead of “to compare the immune response”, the authors could be more specific here, considering in the figure only CD4 and CD8 are being presented. This change would be implemented in the legends of all subsequent figures.
· Figure 4: The legend of the figure says Mann-Whitney test, but in the figure, it says T-test. The T-test was not mentioned in the “statistical analysis” section. Please, update the statistical section (or Figure 4) accordingly.
· Line 274: The authors say 10 subtypes in the introduction and here it says 11.
· Overall, the authors attribute the differences found in the host’s markers to the different infection profiles by KoRV. However, considering the cross-sectional nature of the study and the fact naturally infected animals are used, can we rule out that there are genetically susceptible koalas and that this would impact on the way the virus interact with receptors and further modulate the immune response?
· Likewise, throughout the manuscript, authors use the verb “suppress” to describe the changes found and indicate that the KoRV infection profiles suppress this or that marker. However, although this could be a hypothesis for the change, the assumption cannot be really made based on the present results. Therefore, I suggest the authors describe the correlations without assuming a causation. This hypothesis can of course be raised in the discussion, as long as limitations to the interpretation are presented.
Author Response
Comments and Suggestions for Authors
This is an interesting study that I was pleased to review, as I believe the assessment of the immune response in koalas and wildlife species in general is still a challenge. Thus, contributions to this matter are extremely relevant for the better understanding of the response of these species to pathogens, especially considering their role in the one health context and their often endangered status.
Below are some comments and points I believe need further clarification.
- Is there any reason for not describing the recently identified KoRV subtypes L-M? I understand they wouldn’t be included in the study, but they are also not raised in the introduction.
Response: Thanks for the reviewer comments. The recently identified subtypes L and M were missed as the manuscript was drafted earlier and not searched for any new subtypes. We are grateful to the reviewer for indicating the newly identified subtypes. Accordingly, we have updated the text throughout the manuscript including these subtypes (p1, line 21, 37; p2 line 83, 87, p10, line 560-561).
- Line 23: IFN-γ
Response: Thank you, we have corrected it.
- Lines 48-50: I think this sentence is not really clear. I suggest rewriting it.
Response: In line with the reviewer comment, we have rewritten the sentence (p2, line 71-76).
- Lines 58-59: What do Northern and Southern refer to? Australia?
Response: Thank you. We have clarified in lines 63-64 what we meant for northern and southern (p2, line 85-86).
- The inclusion in methods and results of lots of details that have been already published might not be needed or appropriate. I understand this information needs to be given, but the details could be more concise.
Response: Thanks for the reviewer comments. In line with the reviewer comments, we have deleted few sentences from method section (from subsections Extraction of genomic DNA, Gene expression analysis by quantitative reverse transcription-PCR (qRT-PCR)) to make it more concise (p3-4).
- In the methods, why did the authors decide to use PBMS instead of whole blood for the molecular study of immunological markers?
Response: Thanks for the reviewer comments. We understand that PBMCs do not include all subtypes, but PBMCs play a significant role in the immune system. Actually, as in our previous paper we used PBMC for TLR and cytokine expression, therefore, in this study we also used PBMCs. In addition, from whole blood, it is sometimes very difficult to isolate RNA, possibly due to the RBC to accelerate coagulation.
- Lines 161-162: Could you provide the median of the ages?
Response: We have provided these median (p4 line 197-198).
- Line 163: where it says PBMC, isn’t it whole blood?
Response: Thank you. We have corrected it.
- Line 192: KoRV-A is listed amongst the subtypes but haven’t been analysed.
Response: In line with the reviewer comments, we have included KoRV-A in the study and analyzed and made the figures anew (Figures 2-4).
- Line 197: Instead of “to compare the immune response”, the authors could be more specific here, considering in the figure only CD4 and CD8 are being presented. This change would be implemented in the legends of all subsequent figures.
Response: We have updated the text as per the reviewer comments (p4, line 190; p8, figure legend of Figure 2).
- Figure 4: The legend of the figure says Mann-Whitney test, but in the figure, it says T-test. The T-test was not mentioned in the “statistical analysis” section. Please, update the statistical section (or Figure 4) accordingly.
Response: In response to reviewer 2, to be statistically sound, we have only shown a One-way ANOVA among the groups, and accordingly updated the Statistical analysis section, and we have removed t-test, which was performed by Mann-Whitney test.
- Line 274: The authors say 10 subtypes in the introduction and here it says 11.
Response: We are grateful to the reviewer for his/her careful reading of the manuscript. We have corrected it.
- Overall, the authors attribute the differences found in the host’s markers to the different infection profiles by KoRV. However, considering the cross-sectional nature of the study and the fact naturally infected animals are used, can we rule out that there are genetically susceptible koalas and that this would impact on the way the virus interact with receptors and further modulate the immune response?
Response: We thank the reviewer for his/her comments. We cannot rule out the influence of the genetical susceptibility of koalas, which would affect the host-virus interaction and further modulating the immune response. We have added this information in the discussion section (line 557-559).
- Likewise, throughout the manuscript, authors use the verb “suppress” to describe the changes found and indicate that the KoRV infection profiles suppress this or that marker. However, although this could be a hypothesis for the change, the assumption cannot be really made based on the present results. Therefore, I suggest the authors describe the correlations without assuming a causation. This hypothesis can of course be raised in the discussion, as long as limitations to the interpretation are presented.
Response: We thank the reviewer for his/her comments. According to the reviewer comments we have updated the text in the discussion section (p10, line 547-553; 555-559.
Reviewer 2 Report
The Enamul Hoque Kayesh et al study analyzes the expression level of several immune genes in koalas infected with different types of KoRV. This is an interesting disease, with both endogenous and exogenous types, and there is evidence that is can cause immunosuppression, so it is worth investigation. The acquisition of koala samples is difficult, so anything that can be done with these opportunistically acquired samples from zoos should be published somewhere if the data and the analyses are accurate, and they might be able to add to the understanding of the disease in the literature. That said, the current study is of unclear significance, and there are significant major and minor problems.
Major:
1. There seems to be no mention of a way to correct for multiple comparisons. If a simple Bonferroni correction is made to the Mann-Whitney test, it is unclear if anything would be significant. If some correction was done it should be explained—otherwise the statistical methods are inadequate and the claims are incorrect.
2. If there is any way that the authors could analyze the effect of individual serotypes, controlling for the presence of the other serotypes, that would greatly increase the importance of this paper. As is, there are no conclusions about which serotypes are involved in the responses observed. It may be, for example, based on the groups in Figure 2A, that the presence of type C could independently cause a decrease in CD4 mRNA. IFN-gamma might fit this pattern too, but it is unclear from the data as presented. Even if this cannot be confirmed by this study and it is just preliminary it would allow the study to generate hypotheses that could move the field forward and make testable predictions. That would be more informative than the conclusion that animals with one specific set of types (present in only 5 of 37 animals) has a possibly lower CD4 level than one of the other sets (in only 4 of 37 animals), and that nothing is said of the remaining 28/37 animals. Even if the data are just descriptive (with nothing significant), this type of analysis might make the study more informative. Having a descriptive study with no significant differences is better than one with significant but meaningless differences or with differences that are claimed to be significant through incorrect statistical analysis.
Minor:
1. In the abstract it is unclear what the A+B+C, etc nomenclature is supposed to mean. By reading Table 2, we can understand that this really does represent animals infected with multiple serotypes. This should be clarified in the abstract.
2. The results of the 2 animals with no KoRV B (which were dropped from the analysis) should be mentioned.
3. Are the numbers of animals with each serotype expected given the known prevalences in the wild?
Author Response
Comments and Suggestions for Authors
The Enamul Hoque Kayesh et al study analyzes the expression level of several immune genes in koalas infected with different types of KoRV. This is an interesting disease, with both endogenous and exogenous types, and there is evidence that is can cause immunosuppression, so it is worth investigation. The acquisition of koala samples is difficult, so anything that can be done with these opportunistically acquired samples from zoos should be published somewhere if the data and the analyses are accurate, and they might be able to add to the understanding of the disease in the literature. That said, the current study is of unclear significance, and there are significant major and minor problems.
Major:
- There seems to be no mention of a way to correct for multiple comparisons. If a simple Bonferroni correction is made to the Mann-Whitney test, it is unclear if anything would be significant. If some correction was done it should be explained—otherwise the statistical methods are inadequate and the claims are incorrect.
Response: We are very grateful to the reviewer for his/her kind comments and careful reading of the manuscript. In line with the reviewer comments, we have made correction in statistical analysis, and we have shown only the analysis of One-way ANOVA, which is suitable for multiple comparisons, and we have removed the analysis of Mann-Whitney test that was used for comparison between the groups.
- If there is any way that the authors could analyze the effect of individual serotypes, controlling for the presence of the other serotypes, that would greatly increase the importance of this paper. As is, there are no conclusions about which serotypes are involved in the responses observed. It may be, for example, based on the groups in Figure 2A, that the presence of type C could independently cause a decrease in CD4 mRNA. IFN-gamma might fit this pattern too, but it is unclear from the data as presented. Even if this cannot be confirmed by this study and it is just preliminary it would allow the study to generate hypotheses that could move the field forward and make testable predictions. That would be more informative than the conclusion that animals with one specific set of types (present in only 5 of 37 animals) has a possibly lower CD4 level than one of the other sets (in only 4 of 37 animals), and that nothing is said of the remaining 28/37 animals. Even if the data are just descriptive (with nothing significant), this type of analysis might make the study more informative. Having a descriptive study with no significant differences is better than one with significant but meaningless differences or with differences that are claimed to be significant through incorrect statistical analysis.
Response: In line with the reviewer comments, we have updated statistical analysis and including only one-way ANOVA, which is suitable and correct statistical analysis for multiple group analysis. We have also removed the text of comparison between the groups and their analysis. With these samples which are infected with multiple subtypes, and with the present data, it is not possible to confirm the effects of individual subtypes, because we are not confirmed of remaining other KoRV subtypes in the samples. Moreover, the genetic susceptibility of animals towards immune response cannot be ruled out. Accordingly, we have updated the text in the discussion section (line 548-553; 556-560).
Minor:
- In the abstract it is unclear what the A+B+C, etc nomenclature is supposed to mean. By reading Table 2, we can understand that this really does represent animals infected with multiple serotypes. This should be clarified in the abstract.
Response: We have updated the abstract as per reviewer comments (line 22-23).
- The results of the 2 animals with no KoRV B (which were dropped from the analysis) should be mentioned.
Response: In line with reviewer comments, we have added the analysis and updated the figures (new Figures 2-4).
- Are the numbers of animals with each serotype expected given the known prevalences in the wild?
Response: The prevalence of each serotype (KoRV-A to F) of these samples has been indicated in the recently published paper (Hashem et al., 2022; PMID: 35533919). The overall prevalence of KoRV-A was 100%, and the overall prevalences of KoRV-B, -C, -D, and -F subtypes were 94.59, 27.03, 67.57, and 54.05%, respectively, but KoRV-E was not detected. It differs from the recently reported prevalence of these subtypes in wild northern koalas, where KoRV-D was the most prevalent non-KoRV-A subtype (Blyton et al., 2022; PMID: 35588407). We have also discussed this point in the discussion section (line 541-545).
Round 2
Reviewer 2 Report
The authors have revised the statistical analysis and all current analyses appear appropriate. It is unfortunate that they were not able do any analysis that could attempt to separate the data on individual serotypes. There are several sets of comparison groups (such as the ABD/ABCD and ABDF/ABCDF pairs) that differ in only one serotype that might enable significant conclusions. Something like a simple linear regression model coding each seroptye as present or absent (instead of having each combination of serotypes as a single group) could be useful here. Alternatively, if the data were available, other scientists could provide these analyses. I think this would dramatically increase the significance of the report.
Author Response
Response: In line with reviewer comments, we have separated the data and analyzed separately based on several sets of comparison groups such as AB/ABD _Figure S1, AB/ABF _Figure S2, ABD/ABCD _Figure S3, ABD/ABDF _Figure S4, and ABDF/ABCDF _Figure S5 that differ in only one serotype either KoRV-D, -F, or -C. Accordingly, we have shown in the revised file as supplementary figure S1 to S5, and updated the text (line 279-351).

This manuscript is a resubmission of an earlier submission. The following is a list of the peer review reports and author responses from that submission.
Round 1
Reviewer 1 Report
The study of Hoque Kayesh et al reports the characterization of a certain number of immune parameters in koalas infected with the different KoRVs.
The findings described in this study are of potential interest, however to put infection in relationship to the immune parameters tested here, the quantification of the virus titers/proviral load or must be provided. For the endogenous KoARV-A, levels of RNA could also be quantified.
Author Response
The study of Hoque Kayesh et al reports the characterization of a certain number of immune parameters in koalas infected with the different KoRVs.
The findings described in this study are of potential interest, however to put infection in relationship to the immune parameters tested here, the quantification of the virus titers/proviral load or must be provided. For the endogenous KoARV-A, levels of RNA could also be quantified.
Response: We would like to thank reviewer for his/her careful reading of the manuscript. Proviral load and viral RNA titers were provided in deferent paper of our group [31], therefore we have indicated. The modified Table 2 with KoRV RNA level is as follows.
Table 2. Detailed information of koalas used in this study.
Japanese Zoo |
Koala |
Age (during sampling) |
Sex |
KoRV proviral load(copies/103koala β-actin copies) |
KoRV RNA load (copies/103koala β-actin copies) |
KoRV subtypes |
WBC 102/ml blood |
Health status |
|||||
KoRV-A |
KoRV-B |
KoRV-C |
KoRV-D |
KoRV-E |
KoRV-F |
||||||||
Kanazawa Zoo (n=4)
|
KU_KAZ_01 |
1Y 3M |
F |
3611 |
92 |
P |
P |
N |
N |
ND |
P |
115 |
Healthy |
KU_KAZ_02 |
6Y 8M |
M |
3321 |
11 |
P |
P |
N |
N |
ND |
P |
75.5 |
Healthy |
|
KU_KAZ_3 |
8 Y 1 M |
F |
5789 |
82 |
P |
P |
P |
P |
ND |
N |
66.5 |
Healthy |
|
KU_KAZ_4 |
4 Y 3 M |
F |
4488 |
38 |
P |
P |
N |
P |
ND |
P |
95.5 |
Healthy |
|
Tama Zoological Park (n=2) |
KU_TZ_05 |
6Y |
M |
2959 |
42 |
P |
P |
P |
P |
ND |
N |
- |
Healthy |
KU_TZ_06 |
5Y |
M |
2338 |
42 |
P |
P |
N |
P |
ND |
P |
- |
Healthy |
|
Kobe Oji Zoo (n=8) |
KU_KZ_07 |
11Y |
F |
3234 |
55 |
P |
P |
N |
N |
ND |
P |
45 |
Healthy |
KU_KZ_08 |
7Y 2M |
F |
2472 |
12 |
P |
P |
N |
P |
ND |
P |
54 |
Healthy |
|
KU_KZ_09 |
5Y 5M |
M |
3304 |
19 |
P |
P |
N |
P |
ND |
N |
68 |
Healthy |
|
KU_KZ_10 |
4Y 11M |
F |
2526 |
9 |
P |
N |
N |
P |
ND |
P |
68.5 |
Healthy |
|
KU_KZ_11 |
4Y |
M |
5971 |
71 |
P |
P |
P |
P |
ND |
P |
30 |
Healthy |
|
KU_KZ_12 |
2Y 8M |
F |
2475 |
14 |
P |
P |
N |
P |
ND |
P |
147 |
Healthy |
|
KU_KZ_13 |
2Y 3M |
F |
3287 |
110 |
P |
P |
N |
N |
ND |
N |
- |
Healthy |
|
KU_KZ_14 |
2Y 3M |
F |
2605 |
25 |
P |
P |
N |
N |
ND |
N |
95.5 |
Healthy |
|
Awaji Farm England Hill Zoo (n=4) |
KU_AZ_15 |
24 Y |
F |
4 |
0 |
P |
N |
N |
N |
ND |
N |
50 |
Healthy |
KU_AZ_16 |
13 Y |
F |
13 |
0.01 |
P |
P |
N |
N |
ND |
N |
58 |
Healthy |
|
KU_AZ_17 |
12 Y |
M |
9 |
0 |
P |
P |
N |
P |
ND |
N |
97 |
Healthy |
|
KU_AZ_18 |
7 Y |
M |
3 |
0 |
P |
P |
N |
P |
ND |
P |
42 |
Healthy |
|
Hirakawa Zoological Park (n=4) |
KU_HZ_19 |
1Y |
M |
2797 |
4332 |
P |
P |
P |
P |
ND |
N |
62 |
Healthy |
KU_HZ_20 |
1Y |
M |
2846 |
1220 |
P |
P |
N |
P |
ND |
N |
50 |
Healthy |
|
KU_HZ_21 |
1Y |
F |
3142 |
522 |
P |
P |
N |
P |
ND |
P |
82 |
Healthy |
|
KU_HZ_22 |
1Y |
F |
3872 |
959 |
P |
P |
N |
P |
ND |
N |
88 |
Healthy |
|
Saitama Children`s Zoo (n=5) |
KU_SZ_23 |
3 Y |
M |
6453 |
27 |
P |
P |
N |
P |
ND |
P |
- |
Healthy |
KU_SZ_24 |
7 Y |
F |
4275 |
68 |
P |
P |
P |
P |
ND |
P |
- |
Healthy |
|
KU_SZ_25 |
3 Y |
F |
6038 |
56 |
P |
P |
P |
P |
ND |
P |
- |
Healthy |
|
KU_SZ_26 |
2 Y |
F |
3113 |
168 |
P |
P |
P |
P |
ND |
N |
- |
Healthy |
|
KU_SZ_27 |
2 Y |
F |
4508 |
29 |
P |
P |
P |
P |
ND |
P |
- |
Healthy |
|
Nagoya Higashiyama Zoo (n=10) |
KU_NZ_28 |
9 Y |
M |
3787 |
32 |
P |
P |
P |
P |
ND |
N |
76 |
Healthy |
KU_NZ_29 |
4 Y |
M |
7325 |
35 |
P |
P |
N |
P |
ND |
P |
85 |
Healthy |
|
KU_NZ_30 |
11 Y |
F |
2804 |
33 |
P |
P |
N |
P |
ND |
P |
23 |
Healthy |
|
KU_NZ_31 |
11 Y |
F |
4555 |
17 |
P |
P |
P |
P |
ND |
P |
51 |
Healthy |
|
KU_NZ_32 |
7 Y |
F |
7400 |
17 |
P |
P |
N |
N |
ND |
N |
47 |
Healthy |
|
KU_NZ_33 |
4 Y |
F |
3968 |
8 |
P |
P |
N |
P |
ND |
P |
57 |
Healthy |
|
KU_NZ_34 |
10 M |
F |
5156 |
52 |
P |
P |
N |
N |
ND |
N |
43 |
Healthy |
|
KU_NZ_35 |
1 Y |
F |
4722 |
48 |
P |
P |
N |
N |
ND |
P |
69 |
Healthy |
|
KU_NZ_36 |
11 M |
M |
2953 |
30 |
P |
P |
N |
N |
ND |
N |
69 |
Healthy |
|
KU_NZ_37 |
3 Y |
F |
2808 |
33 |
P |
P |
N |
N |
ND |
N |
66 |
Healthy |
* P, positive; N, negative; M, male or month; F, female; Y, year; ND, not detected; WBC, white blood cell; -, data not available.

Reviewer 2 Report
The effect of KoRV infection to Koala health conditions is currently not clear. In this study, the authors try to clarify the effect from several view points.
The study design was not appropriate.
The authors targeted only on proviral genome to analyze the relationship between immune response and infection status, and the levels of viral RNA, basic information of retrovirus infection, were not collected. Additionally, although the health status of Koalas was also collected, it was not mentioned in the result section. The levels of viral RNA was correlated to the symptoms and that information should be showed for analysis of the relationship between infection status and immune response.
And, the statistical analysis should be performed with expert in statistical analysis. When no significant difference was observed in one-way ANOVA, Mann-Whitney U test should not be performed normally, I think. And, the classification should be reconsidered. So, I strongly recommend the collaboration with statistical expert.
In the analysis, all animals were positive for KoRV-A and KoRV-A negative animal was not included in this study. Therefore, information on KoRV-A infection is not needed, and this group should be removed for all analyses.
And, Table 2 was busy. The abbreviation should be used to keep table clean.
Author Response
The effect of KoRV infection to Koala health conditions is currently not clear. In this study, the authors try to clarify the effect from several view points.
The study design was not appropriate.
The authors targeted only on proviral genome to analyze the relationship between immune response and infection status, and the levels of viral RNA, basic information of retrovirus infection, were not collected.
Response: We would like to thank reviewer for his careful reading of the manuscript. The viral RNA titers in the study population are shown in reference 31, therefore, we have modified the text. The modified Table 2 with KoRV RNA expression level is as follows.
Table 2. Detailed information of koalas used in this study.
Japanese Zoo |
Koala |
Age (during sampling) |
Sex |
KoRV proviral load(copies/103koala β-actin copies) |
KoRV RNA load (copies/103koala β-actin copies) |
KoRV subtypes |
WBC 102/ml blood |
Health status |
|||||
KoRV-A |
KoRV-B |
KoRV-C |
KoRV-D |
KoRV-E |
KoRV-F |
||||||||
Kanazawa Zoo (n=4)
|
KU_KAZ_01 |
1Y 3M |
F |
3611 |
92 |
P |
P |
N |
N |
ND |
P |
115 |
Healthy |
KU_KAZ_02 |
6Y 8M |
M |
3321 |
11 |
P |
P |
N |
N |
ND |
P |
75.5 |
Healthy |
|
KU_KAZ_3 |
8 Y 1 M |
F |
5789 |
82 |
P |
P |
P |
P |
ND |
N |
66.5 |
Healthy |
|
KU_KAZ_4 |
4 Y 3 M |
F |
4488 |
38 |
P |
P |
N |
P |
ND |
P |
95.5 |
Healthy |
|
Tama Zoological Park (n=2) |
KU_TZ_05 |
6Y |
M |
2959 |
42 |
P |
P |
P |
P |
ND |
N |
- |
Healthy |
KU_TZ_06 |
5Y |
M |
2338 |
42 |
P |
P |
N |
P |
ND |
P |
- |
Healthy |
|
Kobe Oji Zoo (n=8) |
KU_KZ_07 |
11Y |
F |
3234 |
55 |
P |
P |
N |
N |
ND |
P |
45 |
Healthy |
KU_KZ_08 |
7Y 2M |
F |
2472 |
12 |
P |
P |
N |
P |
ND |
P |
54 |
Healthy |
|
KU_KZ_09 |
5Y 5M |
M |
3304 |
19 |
P |
P |
N |
P |
ND |
N |
68 |
Healthy |
|
KU_KZ_10 |
4Y 11M |
F |
2526 |
9 |
P |
N |
N |
P |
ND |
P |
68.5 |
Healthy |
|
KU_KZ_11 |
4Y |
M |
5971 |
71 |
P |
P |
P |
P |
ND |
P |
30 |
Healthy |
|
KU_KZ_12 |
2Y 8M |
F |
2475 |
14 |
P |
P |
N |
P |
ND |
P |
147 |
Healthy |
|
KU_KZ_13 |
2Y 3M |
F |
3287 |
110 |
P |
P |
N |
N |
ND |
N |
- |
Healthy |
|
KU_KZ_14 |
2Y 3M |
F |
2605 |
25 |
P |
P |
N |
N |
ND |
N |
95.5 |
Healthy |
|
Awaji Farm England Hill Zoo (n=4) |
KU_AZ_15 |
24 Y |
F |
4 |
0 |
P |
N |
N |
N |
ND |
N |
50 |
Healthy |
KU_AZ_16 |
13 Y |
F |
13 |
0.01 |
P |
P |
N |
N |
ND |
N |
58 |
Healthy |
|
KU_AZ_17 |
12 Y |
M |
9 |
0 |
P |
P |
N |
P |
ND |
N |
97 |
Healthy |
|
KU_AZ_18 |
7 Y |
M |
3 |
0 |
P |
P |
N |
P |
ND |
P |
42 |
Healthy |
|
Hirakawa Zoological Park (n=4) |
KU_HZ_19 |
1Y |
M |
2797 |
4332 |
P |
P |
P |
P |
ND |
N |
62 |
Healthy |
KU_HZ_20 |
1Y |
M |
2846 |
1220 |
P |
P |
N |
P |
ND |
N |
50 |
Healthy |
|
KU_HZ_21 |
1Y |
F |
3142 |
522 |
P |
P |
N |
P |
ND |
P |
82 |
Healthy |
|
KU_HZ_22 |
1Y |
F |
3872 |
959 |
P |
P |
N |
P |
ND |
N |
88 |
Healthy |
|
Saitama Children`s Zoo (n=5) |
KU_SZ_23 |
3 Y |
M |
6453 |
27 |
P |
P |
N |
P |
ND |
P |
- |
Healthy |
KU_SZ_24 |
7 Y |
F |
4275 |
68 |
P |
P |
P |
P |
ND |
P |
- |
Healthy |
|
KU_SZ_25 |
3 Y |
F |
6038 |
56 |
P |
P |
P |
P |
ND |
P |
- |
Healthy |
|
KU_SZ_26 |
2 Y |
F |
3113 |
168 |
P |
P |
P |
P |
ND |
N |
- |
Healthy |
|
KU_SZ_27 |
2 Y |
F |
4508 |
29 |
P |
P |
P |
P |
ND |
P |
- |
Healthy |
|
Nagoya Higashiyama Zoo (n=10) |
KU_NZ_28 |
9 Y |
M |
3787 |
32 |
P |
P |
P |
P |
ND |
N |
76 |
Healthy |
KU_NZ_29 |
4 Y |
M |
7325 |
35 |
P |
P |
N |
P |
ND |
P |
85 |
Healthy |
|
KU_NZ_30 |
11 Y |
F |
2804 |
33 |
P |
P |
N |
P |
ND |
P |
23 |
Healthy |
|
KU_NZ_31 |
11 Y |
F |
4555 |
17 |
P |
P |
P |
P |
ND |
P |
51 |
Healthy |
|
KU_NZ_32 |
7 Y |
F |
7400 |
17 |
P |
P |
N |
N |
ND |
N |
47 |
Healthy |
|
KU_NZ_33 |
4 Y |
F |
3968 |
8 |
P |
P |
N |
P |
ND |
P |
57 |
Healthy |
|
KU_NZ_34 |
10 M |
F |
5156 |
52 |
P |
P |
N |
N |
ND |
N |
43 |
Healthy |
|
KU_NZ_35 |
1 Y |
F |
4722 |
48 |
P |
P |
N |
N |
ND |
P |
69 |
Healthy |
|
KU_NZ_36 |
11 M |
M |
2953 |
30 |
P |
P |
N |
N |
ND |
N |
69 |
Healthy |
|
KU_NZ_37 |
3 Y |
F |
2808 |
33 |
P |
P |
N |
N |
ND |
N |
66 |
Healthy |
* P, positive; N, negative; M, male or month; F, female; Y, year; ND, not detected; WBC, white blood cell; -, data not available.
Additionally, although the health status of Koalas was also collected, it was not mentioned in the result section.
Response: In line with reviewer comments, we have included the statement of health status in the result section (page 5, line).
The levels of viral RNA was correlated to the symptoms and that information should be showed for analysis of the relationship between infection status and immune response.
Response: In line with reviewer comments, we have provided the viral RNA levels in the study population. We also have provided the reference 31 for the effects of multiple subtypes on viral RNA load.
And, the statistical analysis should be performed with expert in statistical analysis. When no significant difference was observed in one-way ANOVA, Mann-Whitney U test should not be performed normally, I think. And, the classification should be reconsidered. So, I strongly recommend the collaboration with statistical expert.
In the analysis, all animals were positive for KoRV-A and KoRV-A negative animal was not included in this study. Therefore, information on KoRV-A infection is not needed, and this group should be removed for all analyses.
Response: In line with reviewer comments, KoRV-A infection group has been removed from statistical analyses. As per statistician comment, we have also removed one more group (KoRV-A+D+F group) due to low number of animals (n=1) in that group. Overall, we have reclassified the animals for analyses (modified figures 2, 3, and 4).
And, Table 2 was busy. The abbreviation should be used to keep table clean.
Response: As per reviewer comments, we have updated the table using abbreviations.

Reviewer 3 Report
In their manuscript “Molecular insight into the innate immune response in captive koala peripheral blood mononuclear cells co-infected with multiple koala retrovirus subtypes“ Kayesh and co-authors correlated the mRNA expression of selected immune response-associated genes (CD4, CD8, cytokines, and TLRs) in PBMCs of captive koalas with the presence of specific combination of KoRV subtypes. All koalas being currently kept in Japan zoos were profiled, which represent an extensive analysis of 37 animals. However, authors divided that cohort into eight categories according to the set of KoRVs (endogenous and exogenous), which makes the statistical analysis difficult because of low numbers of animals in certain categories. Although mRNAs of several immune response-associated genes tested significantly decreased in certain categories, it is difficult to draw any conclusions and the biological reason of this remains vague. Authors stick to the descriptive level and do not bring any valid explanation or hypothesis.
The main obstacle of convincing statistical analysis is the absence of KoRV-free reference cohort of koalas. Authors were aware of this and admitted that they do not know what was the standard gene expression in healthy and uninfected animals.
The presence of KoRVs have been tested by subtype-specific non-quantitative PCR of genomic DNA. Thus, authors did not know either the virus load or the level of KoRV expression because of possible provirus silencing, particularly in the case of the endogenous KoRV-As. Silent proviral copies hardly contributed to the down-regulation of cytokines and TLRs.
Analogically, any suppressive effect of KoRVs on the cytokines or TLRs should be confirmed also at the protein level because the slightly different mRNA levels could have been equalized by more or less efficient translation, protein turnover, etc.
Author Response
In their manuscript “Molecular insight into the innate immune response in captive koala peripheral blood mononuclear cells co-infected with multiple koala retrovirus subtypes“ Kayesh and co-authors correlated the mRNA expression of selected immune response-associated genes (CD4, CD8, cytokines, and TLRs) in PBMCs of captive koalas with the presence of specific combination of KoRV subtypes. All koalas being currently kept in Japan zoos were profiled, which represent an extensive analysis of 37 animals. However, authors divided that cohort into eight categories according to the set of KoRVs (endogenous and exogenous), which makes the statistical analysis difficult because of low numbers of animals in certain categories.
Response: We would like to thank the reviewer for his careful reading of the manuscript and comments. In line with reviewer comments, we have removed KoRV- A and KoRV-A+D+F group from analysis (updated figure 2, 3, 4, and supplementary figure 1)
Although mRNAs of several immune response-associated genes tested significantly decreased in certain categories, it is difficult to draw any conclusions and the biological reason of this remains vague. Authors stick to the descriptive level and do not bring any valid explanation or hypothesis.
Response: We would like to thank the reviewer for his careful reading of the manuscript and comments. In line with reviewer comments, we have removed KoRV- A and KoRV-A+D+F group from analysis (updated figure 2, 3, 4, and supplementary figure 1).
The main obstacle of convincing statistical analysis is the absence of KoRV-free reference cohort of koalas. Authors were aware of this and admitted that they do not know what was the standard gene expression in healthy and uninfected animals.
Response: We are grateful to the reviewer for his kind consideration. Right now, all koalas in Japanese zoo are infected with KoRV. Therefore, it is not easy to characterize KoRV-free koala sample in Japan, but we would like to perform future study with KoRV-free population by collaborative study.
The presence of KoRVs have been tested by subtype-specific non-quantitative PCR of genomic DNA. Thus, authors did not know either the virus load or the level of KoRV expression because of possible provirus silencing, particularly in the case of the endogenous KoRV-As. Silent proviral copies hardly contributed to the down-regulation of cytokines and TLRs.
Response: We would like to thank the reviewer for his comment. We have reported KoRV expression level in reference 31, therefore, we have referred this information in the manuscript. Summarized Table2 with KoRV expression is as follows.
Table 2. Detailed information of koalas used in this study.
Japanese Zoo |
Koala |
Age (during sampling) |
Sex |
KoRV proviral load(copies/103koala β-actin copies) |
KoRV RNA load (copies/103koala β-actin copies) |
KoRV subtypes |
WBC 102/ml blood |
Health status |
|||||
KoRV-A |
KoRV-B |
KoRV-C |
KoRV-D |
KoRV-E |
KoRV-F |
||||||||
Kanazawa Zoo (n=4)
|
KU_KAZ_01 |
1Y 3M |
F |
3611 |
92 |
P |
P |
N |
N |
ND |
P |
115 |
Healthy |
KU_KAZ_02 |
6Y 8M |
M |
3321 |
11 |
P |
P |
N |
N |
ND |
P |
75.5 |
Healthy |
|
KU_KAZ_3 |
8 Y 1 M |
F |
5789 |
82 |
P |
P |
P |
P |
ND |
N |
66.5 |
Healthy |
|
KU_KAZ_4 |
4 Y 3 M |
F |
4488 |
38 |
P |
P |
N |
P |
ND |
P |
95.5 |
Healthy |
|
Tama Zoological Park (n=2) |
KU_TZ_05 |
6Y |
M |
2959 |
42 |
P |
P |
P |
P |
ND |
N |
- |
Healthy |
KU_TZ_06 |
5Y |
M |
2338 |
42 |
P |
P |
N |
P |
ND |
P |
- |
Healthy |
|
Kobe Oji Zoo (n=8) |
KU_KZ_07 |
11Y |
F |
3234 |
55 |
P |
P |
N |
N |
ND |
P |
45 |
Healthy |
KU_KZ_08 |
7Y 2M |
F |
2472 |
12 |
P |
P |
N |
P |
ND |
P |
54 |
Healthy |
|
KU_KZ_09 |
5Y 5M |
M |
3304 |
19 |
P |
P |
N |
P |
ND |
N |
68 |
Healthy |
|
KU_KZ_10 |
4Y 11M |
F |
2526 |
9 |
P |
N |
N |
P |
ND |
P |
68.5 |
Healthy |
|
KU_KZ_11 |
4Y |
M |
5971 |
71 |
P |
P |
P |
P |
ND |
P |
30 |
Healthy |
|
KU_KZ_12 |
2Y 8M |
F |
2475 |
14 |
P |
P |
N |
P |
ND |
P |
147 |
Healthy |
|
KU_KZ_13 |
2Y 3M |
F |
3287 |
110 |
P |
P |
N |
N |
ND |
N |
- |
Healthy |
|
KU_KZ_14 |
2Y 3M |
F |
2605 |
25 |
P |
P |
N |
N |
ND |
N |
95.5 |
Healthy |
|
Awaji Farm England Hill Zoo (n=4) |
KU_AZ_15 |
24 Y |
F |
4 |
0 |
P |
N |
N |
N |
ND |
N |
50 |
Healthy |
KU_AZ_16 |
13 Y |
F |
13 |
0.01 |
P |
P |
N |
N |
ND |
N |
58 |
Healthy |
|
KU_AZ_17 |
12 Y |
M |
9 |
0 |
P |
P |
N |
P |
ND |
N |
97 |
Healthy |
|
KU_AZ_18 |
7 Y |
M |
3 |
0 |
P |
P |
N |
P |
ND |
P |
42 |
Healthy |
|
Hirakawa Zoological Park (n=4) |
KU_HZ_19 |
1Y |
M |
2797 |
4332 |
P |
P |
P |
P |
ND |
N |
62 |
Healthy |
KU_HZ_20 |
1Y |
M |
2846 |
1220 |
P |
P |
N |
P |
ND |
N |
50 |
Healthy |
|
KU_HZ_21 |
1Y |
F |
3142 |
522 |
P |
P |
N |
P |
ND |
P |
82 |
Healthy |
|
KU_HZ_22 |
1Y |
F |
3872 |
959 |
P |
P |
N |
P |
ND |
N |
88 |
Healthy |
|
Saitama Children`s Zoo (n=5) |
KU_SZ_23 |
3 Y |
M |
6453 |
27 |
P |
P |
N |
P |
ND |
P |
- |
Healthy |
KU_SZ_24 |
7 Y |
F |
4275 |
68 |
P |
P |
P |
P |
ND |
P |
- |
Healthy |
|
KU_SZ_25 |
3 Y |
F |
6038 |
56 |
P |
P |
P |
P |
ND |
P |
- |
Healthy |
|
KU_SZ_26 |
2 Y |
F |
3113 |
168 |
P |
P |
P |
P |
ND |
N |
- |
Healthy |
|
KU_SZ_27 |
2 Y |
F |
4508 |
29 |
P |
P |
P |
P |
ND |
P |
- |
Healthy |
|
Nagoya Higashiyama Zoo (n=10) |
KU_NZ_28 |
9 Y |
M |
3787 |
32 |
P |
P |
P |
P |
ND |
N |
76 |
Healthy |
KU_NZ_29 |
4 Y |
M |
7325 |
35 |
P |
P |
N |
P |
ND |
P |
85 |
Healthy |
|
KU_NZ_30 |
11 Y |
F |
2804 |
33 |
P |
P |
N |
P |
ND |
P |
23 |
Healthy |
|
KU_NZ_31 |
11 Y |
F |
4555 |
17 |
P |
P |
P |
P |
ND |
P |
51 |
Healthy |
|
KU_NZ_32 |
7 Y |
F |
7400 |
17 |
P |
P |
N |
N |
ND |
N |
47 |
Healthy |
|
KU_NZ_33 |
4 Y |
F |
3968 |
8 |
P |
P |
N |
P |
ND |
P |
57 |
Healthy |
|
KU_NZ_34 |
10 M |
F |
5156 |
52 |
P |
P |
N |
N |
ND |
N |
43 |
Healthy |
|
KU_NZ_35 |
1 Y |
F |
4722 |
48 |
P |
P |
N |
N |
ND |
P |
69 |
Healthy |
|
KU_NZ_36 |
11 M |
M |
2953 |
30 |
P |
P |
N |
N |
ND |
N |
69 |
Healthy |
|
KU_NZ_37 |
3 Y |
F |
2808 |
33 |
P |
P |
N |
N |
ND |
N |
66 |
Healthy |
* P, positive; N, negative; M, male or month; F, female; Y, year; ND, not detected; WBC, white blood cell; -, data not available.
Analogically, any suppressive effect of KoRVs on the cytokines or TLRs should be confirmed also at the protein level because the slightly different mRNA levels could have been equalized by more or less efficient translation, protein turnover, etc.
Response: We would like to thank the reviewer for his evaluation of the study and for his sincere comment. We would like to continuing further investigation in the future study to develop analytical tool for confirming koala TLR and cytokines protein expression by WB, as we do not have enough amount of koala samples to characterize protein level right now.

Round 2
Reviewer 1 Report
The authors have provided some information on virus replication rates in the table 2, but the current table appended in the revised form seems still the old one.
Also, it is not sufficient to attach the information but it is also needed to study the relationship -or lack thereof- between this novel parameter and the immunological ones measured in this study. As such, the authors have not adequately addressed the comments of this reviewer
Reviewer 2 Report
Statistical analysis is not performed using appropriate methods. In Figure 2, for example, group comparisons were performed even though one-way ANOVA showed no significant difference, or nonparametric and parametric tests were mixed. A discussion based on results that have not been properly analyzed would be meaningless.
Reviewer 3 Report
In their revised manuscript, authors did very few to cope with my criticism. The n=1 cohorts of KoRV-A- and KoRV-A+D+F-infected animals were excluded, which slightly improved the statistical analysis. However, I am still confused with the absence of clear conclusions and biological explanation of the findings. Without the cohort of uninfected animals as a negative control (which is difficult to obtain), what could be said e.g. about the importance of KoRV subtypes? Would principal component analysis help to draw any convincing conclusion?
Authors added proviral loads and RNA loads to the Table 2. These data is, however, not subtype-specific and does not help very much in explaining the subtle differences in expression of the CD4, CD8, cytokines, and TLRs. Interesting is the marginal proviral load and zero KoRV RNA level in animals (just four, unfortunately) kept in Awaji Farm (although authors write that „there were no significant differences in the KoRV proviral and KoRV RNA loads“, lines 169-170). How about using this cohort instead of the missing negative control? In contrast, four koalas from the Hirakawa Zoological Park displayed strikingly high RNA load.
I am still unsatisfied that the paper is rather descriptive and without valid explanations or hypotheses.